# Sugar Reduction in Yogurt Products Sold in the UK between 2016 and 2019

**DOI:** 10.3390/nu12010171

**Published:** 2020-01-08

**Authors:** J. Bernadette Moore, Eiméar H. Sutton, Neil Hancock

**Affiliations:** School of Food Science and Nutrition, University of Leeds, Leeds LS2 9JT, UK; eimearsutton97@gmail.com (E.H.S.); N.Hancock@leeds.ac.uk (N.H.)

**Keywords:** yogurt, sugar, childhood obesity

## Abstract

The UK government has called for industry reformulation of foods that contribute most to sugar consumption in children’s diets, including yogurts. The aim of this work was to comprehensively survey yogurt products available in UK supermarkets in 2019 to determine whether sugar contents had been reduced since our baseline survey in 2016. Product information was collected for 893 unique yogurt, fromage frais and dairy dessert products, and nutrient contents were analysed in comparison to those previously examined. Examining all products, there was a highly significant (*p* < 0.0001) reduction in the median total sugar contents in 2019 compared to those in 2016: median (interquartile range): 10.4 g/100 g (6.6, 13.0) versus 11.9 g/100 (8.8, 13.6). However, notable product turnover was evident; while 60% of the 2019 products surveyed could be matched by brand and name to 2016, 40% were new. In scrutinising paired products closely, only 32% (173 of 539) had reduced sugar contents with a smaller mean difference of −0.65 g/100 g (*p* < 0.0001), suggesting that the overall median had dropped as a result of higher sugar products being discontinued. Categories showing the most improvements were children’s, drinks and fruit yogurts. Although only 15% of the 2019 products contained ≤5 g/100 g sugars, considered a ‘low-sugar’ product for labelling, this was an improvement over the 9% identified in 2016. Our results yield important insights into current market trends and demonstrate that the median sugar content of UK yogurt products has been reduced by 13% in two years. These data independently evidence modest, but encouraging changes in response to public policy initiatives aimed at preventing childhood obesity.

## 1. Introduction

Yogurt is a fermented milk product and nutrient-dense food with its unique food matrix influencing the bioavailability of nutrients. It is an easily digestible source of protein and multiple essential micronutrients including calcium and vitamin B12 [1]. There are a variety of identifiable health benefits from consuming fermented dairy products, possibly derived in part from the potential for the probiotic bacteria to colonise the intestinal tract and influence health [2]. However, the final nutrient composition of yogurt will depend on the type of milk used and the ingredients added during production, which often include additional sugars and other sweetening agents [3]. As yogurt is a commonly recommended and consumed food in childhood, sweetened yogurts are a significant source of free sugars (see Table 1 for definitions of free, added, and total sugars) for children in many countries [4,5].

It is now well established that the excess consumption of free sugars increases the risk of obesity and dental caries. Therefore, in addition to the World Health Organization (WHO), multiple government agencies around the world have strongly recommended that less than 10% of total energy intake should come from free sugars [6,7]. Current expert consensus is that children under two should consume no added/free sugars [8]. Nonetheless, yogurt, often a staple part of children’s diets, has been found to contribute a significant amount to the free sugar intakes of young children. In an Australian cohort study (*n* = 1043), yogurt accounted for 8.8% of free sugars intake to the diets of 1–2 year olds [5]. Similarly in the UK, yogurt contributed 11.1% of free sugars intakes to children under three, and 5.5% to children aged 4–10 years old [4]. Concerningly, in England in 2017, 30% of children aged 2–15 were overweight or obese, including 17% who were obese [9]. As childhood obesity is a predictor of adult obesity and obesity-related disorders [10], the reduction of free sugars intake is a pragmatic public health message in the context of a portfolio of obesity-prevention strategies [6].

With this motivation, in 2016 the UK government introduced a multipronged action plan to combat childhood obesity [16]. Among other actions, this included a soft-drink-industry levy and a structured reformulation programme aiming for a 20% reduction in the sugar contents of the top nine food categories contributing to sugar in children’s diets, including yogurt and fromage frais products, by 2020. These recommendations prompted our interest in the sugar and nutrient contents of yogurt, and in 2016 we undertook a comprehensive baseline survey of products on the UK yogurt market [15]. Indeed, the sugar contents, particularly in children’s and organic products, were found to be higher than might be expected for a perceived healthy food. With the exception of unsweetened natural/Greek yogurts, the majority of products had high median total sugar contents well above 10 g/100 g accounting for >45% of energy. Only 2% of children’s products met the 5 g/100 g maximum required for a low-sugar claim under EU regulations and a green traffic light label in the UK [17,18].

Reducing the sugar contents of yogurt and dairy desserts by half could reduce sugar consumption in children aged 4–10 by approximately 3.1% [19,20]. If achieved in combination with sugar reduction in other categories within the UK programme, recent modelling suggests the resulting relatively small reduction in energy intake (25 kcal/day for children) could significantly reduce the number of childhood obesity cases with significant cost savings (£286 m over 10 years) in the long-term associated health burdens in adults [21]. These data highlight the impact product reformulations could have, particularly when done in combination. While a recent interim government report suggests that the yogurt and fromage frais category has achieved the highest sugar reduction out of all the categories, with a reduction of 10.3% sales-weighted average (SWA) total sugar level per 100 g [22]; these data have limitations as they are based on consumer panel research and SWA is difficult to interpret at a product level. Therefore, the purpose of this study was to reassess the nutrient contents of yogurt products available in UK supermarkets in 2019 to determine whether the total sugars have been reduced since 2016, and to examine for changes in energy and other macronutrients. 

## 2. Materials and Methods 

### 2.1. Data Collection

Following the same methodology as in the initial survey [15], the terms ‘yogurt’ and ‘yoghurt’ were used to search the online websites of the five major UK supermarket’s (Asda, Morrisons, Sainsbury’s, Tesco, and Waitrose) representing 73% of the UK grocery market share [23] in January 2019. In four out of five cases the change in terms did not alter the number of products presented, although for Tesco more products were shown from the ‘Yoghurt’ search (*n* = 407) than for the ‘Yogurt’ search (*n* = 372), and the larger list was used. All brand and product names presented from each search were collected to make five lists, one for each supermarket. As previously done in November 2016, products such as protein drinks that did not fit within the main food group ‘15: yogurt, fromage frais, and other dairy desserts’ definition of yogurt used in the UK National Diet and Nutrition Survey dietary surveys [24] were deemed out of scope and removed. The five lists were combined, and duplicates were removed to produce one final product list. Duplications were due to the same products being sold in different supermarkets, products being sold in different sizes, or in multipacks as well as single packs. Deduplication was carried out first by using the ‘remove duplicates’ function in Microsoft Excel and then manually due to the complex variety of duplications. 

To compare the new product list with the list from 2016, the Microsoft Excel Fuzzy Lookup Add-In function was used. When the two lists were placed side by side, the programme produced a number indicating the similarity between product names, where number 1 was an exact match and 0 was no similarity. While the programme was useful for indicating exact matches, it did not pick up on products that were the same but had slight variations in naming due to differences in recording or product alterations. Therefore, manual curation was also required to produce three lists: products present in both years, products only in 2016, and products only in 2019. Each product was then given a unique code in order to simplify data analysis and products sold in both years had two codes. In establishing paired products (i.e., the products present in both years), 1.2% (*n* = 11) of products sold both years were identified to have been miscategorised in 2016. Therefore, these products were assigned to the correct categories for 2019 and analysed within these and as part of the 539 paired products.

A database for 2019 was created including the products present in both years and products only in 2019. Each product was placed into one of eight categories using the same systematic process flow methodology used in the original study [15]: children’s, drinks, dairy alternatives, organic, natural/Greek, fruit, flavoured, desserts. As detailed previously, these product groupings and the process flow strategy was formulated a priori driven in part by our interest in examining products marketed to children and based, in large part, on typical groupings used by supermarkets for both inventory and marketing reasons. For example, children’s products were defined as such if either the supermarket or product itself defined them as such; or if cartoons, toy giveaways, games, kids’ clubs, spokes-characters, or celebrities were incorporated into the brand image. These categories give useful added resolution to the striking differences in total sugar and other nutrient contents found among the 893 different products that would be aggregated within the large umbrella food grouping of ‘yogurt, fromage frais, and other dairy desserts’ found in dietary surveys such as the UK National Dietary and Nutrition Survey (NDNS) [15]. In February 2019, food label data per 100 g for energy, fat, saturated fat, carbohydrates, total sugar (as defined in Table 1 this is what is required and available on nutrition labels in the UK), fibre, protein, and calcium were collected from supermarket or brand websites, as well as product portion sizes and ingredient lists. Finally, 10% of the data underwent a spot check to ensure accuracy before data analysis took place. Both the 2019 database [25] developed for this project and the 2016 database [26] are available through an open access repository under a Creative Commons Attribution licence (CC-BY 4.0).

### 2.2. Data Analysis 

Microsoft Excel (Microsoft, Reading, UK) was used for constructing and manipulating the product database, while statistical analyses were carried out using GraphPad Prism V/7.0c (GraphPad Software, San Diego, USA). Data were tested for normality using Prism’s recommended D’Agostino and Pearson omnibus K2 test. To compare the difference between paired products, either the non-parametric Wilcoxon matched-pairs signed-rank test or a paired t-test were carried out dependent on results of the normality test. For overall comparison of 2016 to 2019, either a non-parametric Mann–Whitney or an unpaired t-test were applied. For analysis across categories within 2019 only, the non-parametric Kruskal–Wallis test along with Dunn’s multiple comparisons were used.

## 3. Results

### 3.1. Comparison of Yogurt Products Sold in 2016 and 2019

The initial search across the five supermarket websites returned over 2000 products. Once extraneous and duplicated products were removed, the final 2019 database had 893 unique yogurt, fromage frais, and dairy dessert products in line with the 898 scrutinised in 2016 [15]. Of these, 539 (60.4%; Figure 1a) yogurts were found in common, having the same brand and name in both the 2016 and 2019 databases; while 354 products were new, demonstrating dynamic turnover in available yogurt products during the 26 months between surveys. Although the overall number of products was very similar across the two years, the proportions of the categories within had changed somewhat. For example, although the dairy alternatives category had the fewest number of products (*n* = 38; 4.2%) in 2016, almost double this number of products were identified in 2019 (*n* = 67; 7.5%). Interestingly, the number of products in both the children’s and organic categories, scrutinised for their sugar contents in 2016, were reduced by 23% (*n* = 101 in 2016 to *n* = 78 in 2019) and 27% (*n* = 71 to *n* = 52), respectively. 

When all products were compared, there was a highly significant (*p* < 0.0001) reduction in the median total sugar contents in 2019 compared to 2016 (median (interquartile range): 10.4 g/100 g (6.6, 13.0) in 2019 versus 11.9 g/100 g (8.8, 13.6) in 2016; Figure 1b). Similarly, in comparing the 539 paired products, the total sugar contents in 2019 was significantly lower than that in 2016 with an overall mean difference (MD) (95% confidence interval (CI)) of −0.65 g/100 g (−0.78, −0.52) (*p* < 0.0001; Figure 1c). Although 32% (173 of 539) of the paired products had reduced their sugar contents, the majority (328; 61%) had stayed the same and a minority (38; 7%) appeared to have increased sugar levels (Figure 1c). Among those products that reduced sugar, a broad range of reductions were observed, ranging from −0.1 to −9.8 g/100 g. Whereas the range of observed increases was relatively small ranging from 0.1 to 2.3 g/100 g. While no differences in protein or fat contents were observed between the paired 2019 and 2016 products; there were reductions in carbohydrate (−0.76 g/100 g (−0.94, −0.57); *p* < 0.0001) and energy (−2.24 kcal (−2.80, −1.67); *p* < 0.0001). Importantly, this suggests manufacturers were able to reduce sugar contents without compensating by adding other energetic ingredients.

### 3.2. Changes in Sugar Contents of Yogurts within Different Product Categories

Scrutinising the 539 products sold in both years within their eight categories, we found significant reductions in sugar contents in all categories except for dairy alternatives (Figure 2). The three categories showing the most improvements were children’s, drinks and fruit yogurts. Within the children’s category 61% (*n* = 28 of 46) of products had reduced sugar since 2016, resulting in an MD (CI) of −0.84 g/100 g (−1.10, −0.58) (*p* < 0.0001; Figure 2a). The majority of the products that had made reductions, (26 of 28) had reduced their sugar contents by over 5%, as recommended for the first year of the UK sugar reduction programme. The dairy alternatives category was the only category where it appeared there was a net increase in sugar contents among the 39 paired products, 0.12 (−0.03, 0.27), however this was not a significant difference (*p* = 0.0938; Figure 2b). 

Although dairy desserts are not always yogurts, we included them in our initial 2016 survey because within the UK National Diet and Nutrition Survey they are included in the main food group ‘15: yogurt, fromage frais, and other dairy desserts’ [24] and are also targeted by the sugar reduction programme as ‘puddings’ [27]; therefore, we followed them up here. Of the 77 dessert products paired from both surveys, 18 (23%) had reduced in sugar, with 10 products reducing by ≥10%, resulting in a significant MD of −0.45 (−0.73, −0.18) (*p* = 0.0014; Figure 2c). Notably, the drinks category achieved the largest mean reduction of −1.13 g/100 g (−1.96, −0.29), in spite of only 13 of 40 (33%) products making changes, because several products had made large reductions of >5 g/100 g (*p* = 0.0002; Figure 2d). Although reductions were smaller in magnitude for the 28% (17 of 60) of yogurts within the flavoured product category that made changes, nonetheless a significant mean reduction was achieved −0.49 (−0.74, −0.23) overall (*p* = 0.0003; Figure 2e). The fruit category was the largest category and 43% (73 of 195) of its products had reduced in sugar since 2016 with an MD of −0.95 g/100 g (−1.2, −0.69) (*p* < 0.0001; Figure 2f). As natural/Greek yogurts are by definition unsweetened/unflavoured, it was unsurprising that the range of differences observed in the natural/Greek category were very small, with most products remaining the same (Figure 2g). However, some reductions had been made, with 10 of 47 (21%) products reporting lower sugar contents, resulting in a small MD between years of −0.12 (−0.24, −0.01) that was nonetheless significant (*p* = 0.0379; Figure 2g). Lastly, among the organic products found available in both years, 37% (13 of 35) reduced sugar with an overall MD of −0.48 (−0.79, −0.18) (*p* = 0.0029; Figure 2h).

### 3.3. Total Sugar Contents of 2019 Yogurt Products across Categories

Examining the 2019 products in isolation, significant differences in median total sugars were identified across the eight categories (Figure 3; *p* < 0.0001). The range of sugar contents varied both within and between categories. Predictably, desserts had the highest median and broadest range of total sugar contents 16.3 g/100 g (2.3, 32.9); whereas natural/Greek yogurts had the lowest 4.7 g/100 g (0, 10.9). Median total sugar contents were similar between the children’s (10 g/100 g (0.8, 14.90), flavoured (11.9 g/100 g (3.1, 18.0)), fruit (11.0 g/100 g (03.4, 18.0)), and organic (10.8 g/100 g (4.3, 16.3)) categories (Figure 3). These were all higher than both dairy alternatives (8.1 g/100 g (0, 20.0)) and drinks (5.6 g/100 g (0.1, 15.6)), as well as the natural/Greek yogurts. While the median total sugar contents of children’s and organic products highlighted as high in the previous survey had reduced from 2016 (children’s: 10.0 vs. 10.8 g/100 g; organic: 10.8 vs. 13.1 g/100 g), nonetheless this remains double that required for an on-pack low-sugar claim [17,18]. Although only 15% of the 2019 products contained less than or equal to 5 g/100 g sugars, considered a ‘low-sugar’ product, this was an improvement over the 9% identified in 2016. Only 5% (4 of 78) of products within the children’s category qualified as ‘low sugar’, up only slightly from the 2 of 101 identified in 2016.

### 3.4. Examination of Organic Yogurts Subdivided by Category

Lastly, in the previous survey, the organic category was found to have perhaps surprisingly high sugar contents in comparison to all other categories other than desserts. In recognising the underlying heterogeneity of this category (where products are placed based on a label related to food production). We chose to further examine the sugar contents of the underlying sub-categories in comparison to their corresponding non-organic group. Interesting, in spite of much fewer products in the organic category, nonetheless both organic fruit (Figure 4a) and organic natural/Greek (Figure 4b) yogurts were found to have higher median sugar contents than their non-organic counterparts (*p* = 0.0001 and *p* = 0.024, respectively). On the other hand, no differences in median sugar levels were observed for the flavoured (*p* = 0.1223; Figure 4c) and dessert (*p* = 0.7305; Figure 4d) categories. Although notably the number of organic dessert products was very small (*n* = 3 vs. 150).

## 4. Discussion

In this study, we comprehensively evaluated the sugar and macronutrient contents of 893 yogurt products sold in the UK in 2019 in comparison to products sold in 2016. Our results demonstrate that the median sugar content of yogurt products (all products compared) has been reduced by 13% in this timespan and yield insights into current market trends. In addition, the data highlight the potential efficacy of public policy measures for prompting market change and improving the nutrient profile of commonly consumed foods. 

Recent research has shown that the vast majority of parents significantly underestimate the sugar content of common food items in children’s diets and the degree of parental underestimation of sugar was associated with risk of overweight and obesity [28]. Notably, of the six foods tested in the study, yogurt was the food the majority of parents (92% of 305) underestimated the most, by “seven sugar cubes” [28]. The authors attribute this to the ‘health halo effect’, whereby the sugar contents of perceived healthy foods are underestimated. These data illustrate both a common lack of awareness of the sugar contents of food and potential detrimental effects on children’s health that could potentially be countered through comprehensive product reformulation for sugar reduction. While our study shows that the children’s yogurt category did have one of the largest sugar reductions out of all the categories included, nonetheless the median sugar levels of children’s products was still 10 g/100 g and only 5% of children’s products in 2019 reached the requirement for a ‘low-sugar’ label. This is of concern as many healthy eating guidelines, such as the Change4Life campaign, suggest swapping lower-sugar yogurts in place of desserts or puddings [29]. Yet choices for low-sugar products remain limited, representing only 15% of all the products surveyed and only 4 out of 78 children’s products. Low-sugar yogurt products were largely found in the natural/Greek and dairy alternative categories or were drinks. Hence, further reductions or removal of the children’s yogurts contributing the most sugar in particular is likely warranted, to ensure that when parents make suggested ‘swaps’ they are, in fact, making a healthy choice for their child.

In our original survey organic yogurts had the highest median sugar levels outside of the dessert category [15]. This was a likely surprising finding to many consumers who largely associate organic labels as meaning “healthy” and with lower fat or fewer calories; in spite of the fact that organic is a claim regarding production method and not an implication of either the healthfulness or energy contents [30,31]. Notably, the organic products surveyed were second only to desert products in terms of their fat contents. Without compositional analysis it cannot be ruled out that organic products may use a higher proportion of milk with its intrinsic lactose and galactose from lactose fermentation contributing to the higher total sugar contents. However, examining the ingredients lists of the organic yogurt products shows that many of the products highest in sugar had multiple added sweeteners (sugar, honey, fruit purees, fruit juice concentrates along with fruit) and cream was a commonly used ingredient in products highest in fat (>5 g/100 g) [26]. While the median total sugar contents of all organic products had come down significantly since 2016 (10.8 versus 13.1 g/100 g in 2016), in examining the sub-categories within the organic products, e.g., fruit and natural/Greek products, we show that their median sugar contents were nonetheless higher than those of their non-organic category equivalents. Interestingly, the mean difference of the 35 paired organic products sold in both years was smaller (−0.48 g/100 g) than the difference observed between the medians of all the 71 (2016) and 52 (2019) products surveyed (−2.3 g/100 g) suggesting that the overall median had dropped as a result of higher-sugar products being discontinued. 

Indeed, this study highlights just how dynamic the marketplace for food products is and yields interesting insights into current market and consumer trends. For example, off-setting the observed decreases in the number of children’s and organic products, 23% and 27%, respectively, was an almost two-fold increase (76%) in the number of dairy alternative products (*n* = 67 vs. *n* = 38 in 2016). Dairy alternative and plant-based ‘milks’ have recently gained a lot of consumer interest for a number of health, ethical, and environmental reasons [32]. In response to this, the market has expanded from soya and rice milk, to milks derived from a larger variety of ingredients such as almond, cashew, oat, coconut, and hemp [33]. This trend has been picked up on and applied to yogurt products, leading to the development of products that use almond and cashew nuts as their base, as well as an increased number of soya- and coconut-based yogurts. The question of whether plant-based drinks provide the same nutritional and health benefits as cow’s milk remains under investigation [33,34]. Indeed, as dairy alternatives contain no intrinsic lactose, their total sugar content is derived entirely from added sweeteners. The products surveyed here were quite variable in their total sugar contents. While 37% contained <5 g/100, 27% contained >10 g/100 g total sugar; and 20% of dairy alternative products listed sugar as the second most abundant ingredient after water. Arguably, cow’s milk and plant-based drinks are not nutritionally comparable foods [35]. 

A related market trend was the marked increase in the number of kefir products; from only three kefir products in 2016 to 24 in 2019, a remarkable eight-fold increase. Kefir is a traditional fermented milk product made from a unique starter culture of kefir ‘grains’ comprised of a water-soluble polysaccharide kefiran that encapsulates a complex microbiota of lactic acid and acetic acid bacteria and yeasts [36]. Considered a symbiotic, a combination of both a probiotic and a prebiotic, kefir may convey health benefits through enhancing the growth of beneficial microorganisms in the gut potentially preventing gut dysbiosis associated with a variety of chronic metabolic diseases [37]. In January of 2019, Public Health England published a supplementary report to the sugar reduction guidelines encouraging industry to reduce the sugar in fermented yogurt drinks (including kefir, lassis, and drinks with either disease risk reduction claims, including plant stanols and sterols, or functional health claims) by 20% by 2021 [38]. We had made the observations in our previous report that multiple products with added plant stanols marketed for their cholesterol-lowering merits were extremely high in sugar and that given the data linking high sugar consumption to high cholesterol levels these in particular should be considered for reformulation [15]. In this regard, it was gratifying to observe that among the top 20 products making the largest reductions in sugar content (−4.7 to −9.8 g/100 g) were 15 products with added plant stanols. Although there was some variation in the sugar content of the 24 kefir products examined here, on the whole they had lower total sugar than most sweetened-yogurt products with an average of 5.4 g/100 g. 

While our findings are encouraging, nonetheless there are some limitations to this study. Firstly, the survey is only a snapshot of the market at a particular time. As evident from the extent of market changes between 2016 and 2019, it is likely that even within the months of writing this report some products may have been removed and some added, particularly seasonal products. Data were collected from nutrition and ingredient food label information found on supermarket and brand websites that should be up to date but may not have been. Moreover, food labels have limitations. Although in the UK and EU ingredients must be listed in order of weight, with the main ingredient first, manufacturers are not required to list the g/100 g amounts of added sugars on labels and only very few manufactures state the % of milk or sugar or fruit they use. In the absence of such quantitative information, precisely determining the changes in formulation in the 32% (173 of 539) of paired products that had lowered their sugar contents was not possible. Caution may be warranted as the reformulation of products, whether driven by policy initiatives or in response to consumer demand, can have unintended consequences, such as substituting one unhealthy ingredient for another [39]. 

Relatedly, because we did not analytically measure the sugar contents of the 893 products surveyed, it was not possible to assess the free sugar contents separately from lactose, the intrinsic milk sugar excluded from the definition of free sugar. Although there is consensus that in relation to health risks associated with dietary sugar, public guidance and intake monitoring should focus on free sugars [40]; in the UK and most countries globally, only total sugar is reported on nutrition labels and the UK’s sugar reduction guidelines accordingly focus on total sugars. While the reporting of added sugars (under total sugars) on food labels has been recently mandated in the US [13], notably the US Food and Drug Administration’s definition of added sugars excludes fruit puree, a common reason for a high free sugar content in children’s yogurts and fromage frais. While fruit puree may have benefits (e.g., polyphenols) over cane sugar, nonetheless the sugars in fruit puree and ‘natural’ fruit juice concentrates have the same detriments in terms of dental caries and the risk of positive energy balance. Concerningly, fruit purees have recently been touted by some in the food industry as a strategy for circumventing the “added sugar label hurdle” [41]. In the UK products surveyed here approximately 13% (122 of 893) contained one or more fruit purees as ingredients. 

Determining the impact of policy initiatives aimed at reducing sugar consumption will ultimately require appraising purchasing behaviour and sales overtime. Substitution and displacement effects in response to food policies can be unpredictable [39]. Nonetheless, assessments like this one are an important way to capture and expose the actual nutrient profile of foods consumed regularly by adults and children. In the context of an epidemic of childhood obesity and adult cardiometabolic diseases, these data independently evidence encouraging changes in the food environment in response to a government-led, structured reformulation programme. A strength to this work was the large number of products analysed from five supermarkets representing 73% of the UK grocery market [23]. Moreover, the continuity between the method carried out in the previous study is a significant strength, resulting in an informative assessment of reductions made in the sugar contents of yogurt products in the UK market over time.

## 5. Conclusions

In this comprehensive survey of the UK yogurt market, we demonstrate that the median sugar contents of all yogurt products available has been reduced since our 2016 survey. Categories showing the most improvements were children’s, drinks, and fruit yogurts. Fifteen percent of the 2019 products contained less than or equal to 5 g/100 g sugars, considered a ‘low-sugar’ product, an improvement over the 9% identified in 2016. Nonetheless in comparing 539 paired products between the two surveys, 61% had made no changes; suggesting that the overall median had dropped largely as a result of higher-sugar products being discontinued. While, a larger reduction in most categories is needed in order to reach the recommended 20% reduction by 2020; we conclude the sugar content of UK yogurt products has reduced since the sugar reduction programme was put into place in 2016. Significantly, the data evidence modest, but positive changes in the food environment in response to public policy initiatives aimed at preventing childhood obesity.

## Figures and Tables

**Figure 1 nutrients-12-00171-f001:**
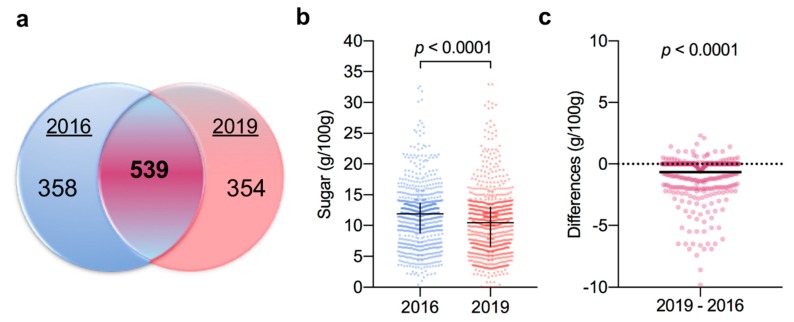
Comparison of total sugar content in yogurt products sold in 2016 and 2019. (**a**) Analysis of product names and brands identifies 539 products sold in both years. (**b**) Distribution of sugar content of all products from 2016 (*n* = 898) and 2019 (*n* = 893). Data were analysed using the Mann–Whitney test; black cross indicates median and interquartile range. (**c**) Difference plot of the change in sugar content between the products sold both in 2019 and 2016 (*n* = 539). Data were analysed using the Wilcoxon matched-pairs signed-rank test; solid black line indicates mean change (−0.6113) between 2016 and 2019.

**Figure 2 nutrients-12-00171-f002:**
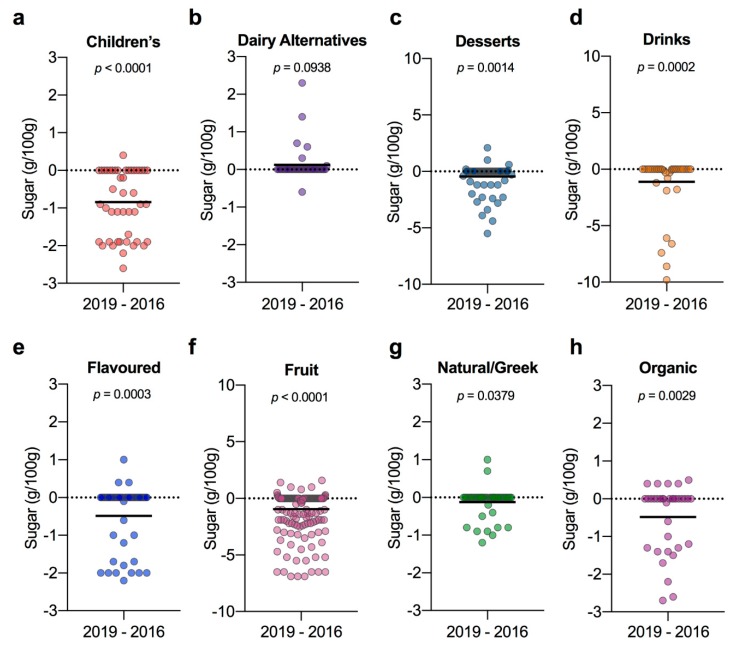
Difference plots for each category showing the change in total sugar content between yogurt products sold both in 2019 and 2016. (**a**) Children’s (*n* = 46); (**b**) dairy alternatives (*n* = 39); (**c**) desserts (*n* = 77); (**d**) drinks (*n* = 40); (**e**) flavoured (*n* = 60); (**f**) fruit (*n* = 195); (**g**) natural/Greek (*n* = 47); (**h**) organic (*n* = 35). Data were analysed using the Wilcoxon matched-pairs signed-rank test; solid black line indicates mean change between 2016 and 2019.

**Figure 3 nutrients-12-00171-f003:**
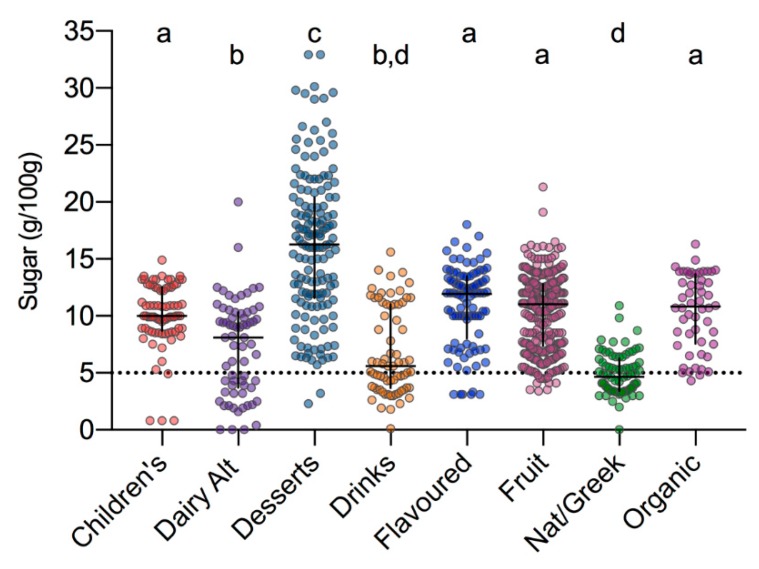
Total sugar content of UK yogurt products across categories. Black cross indicates median and interquartile range. Data were tested for normality and analysed using the Kruskal–Wallis and Dunn’s multiple comparison tests; categories not assigned the same lettering (a–d) are significantly different. Dashed line indicates threshold defined by European Union (EU) regulations for low-sugar nutrition claim [17].

**Figure 4 nutrients-12-00171-f004:**
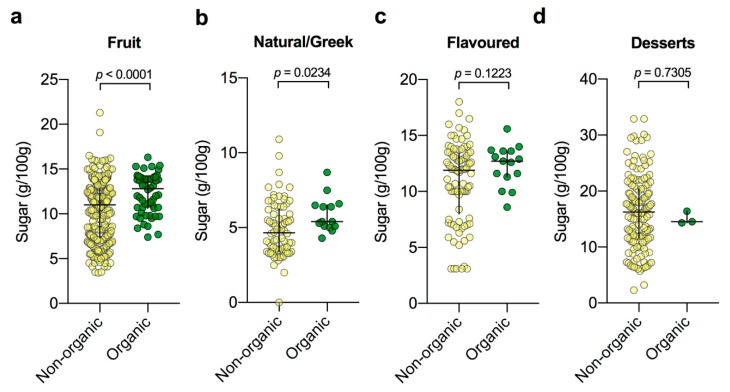
Sugar contents of organic yogurts subdivided by category in comparison to non-organic yogurts. (**a**) Fruit non-organic (*n* = 305) vs. organic (*n* = 56); (**b**) natural/Greek non-organic (*n* = 78) vs. organic (*n* = 14); (**c**) flavoured non-organic (*n* = 93) vs. organic (*n* = 15); (**d**) dessert non-organic (*n* = 150) vs. organic (*n* = 3). Data were tested for normality and analysed appropriately using either the Mann–Whitney or unpaired t-test. The black cross indicates median and interquartile range.

**Table 1 nutrients-12-00171-t001:** Definitions.

Term	Definition
Sugars	Conventionally describes chemically the monosaccharides (glucose, fructose, galactose) and disaccharides (sucrose, lactose, maltose). Sugars includes those occurring naturally in foods and drinks or added during processing and preparation.
Free Sugars	‘All monosaccharides and disaccharides added to foods by the manufacturer, cook, or consumer, plus sugars naturally present in honey, fruit juices, and syrups’ [11]. Under this definition, sugars present in intact fruits and vegetables and lactose naturally present in milk and milk products are excluded.
Total Sugars	Currently required for UK nutrition labels. Includes sugars occurring naturally in foods and beverages and those added during processing and preparation.
Added Sugars	A term used in the United States that excludes sugars in juiced or pureed fruits and vegetables that are included in WHO and UK adopted definition of free sugars. ‘Syrups and other caloric sweeteners used as a sweetener in other food products. Naturally occurring sugars such as those in fruit or milk are not added sugars’ [12]. Will be a required subline under ‘total sugars’ for US food labels from 2020 [13].
Lactose	A disaccharide of glucose and galactose. It is often called ‘milk sugar’ because 100% of ‘total sugars’ in milk are lactose. In natural/Greek yogurt ~80% of the sugar is lactose, with the remainder being galactose generated from lactose fermentation [14].

Adapted with permission from Moore et al. [15].

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
