# Peer review of "Sugar Reduction in Yogurt Products Sold in the UK between 2016 and 2019"

_nutrients, 2020, doi:10.3390/nu12010171_

Round 1

Reviewer 1 Report

The study entitled “Sugar reduction in yogurt products sold in the UK between 2016 and 2019” by Moore et al., aimed to evaluate the sugar content of yogurt products existing in the supermarket between the years 2016 and 2019. Among the total number of products, 15% of the 2019 producers contain ≤5g/100g sugar content, which is the UK government's reformulated sugar content levels in 2016. The results seem promising and the article is written well. However, few minor concerns need to be taken care of.

Specify in the text/figure legend whether 1.2% (n=11) are included in the total number of new samples (n= 354; 2019) [Page number 3, line no. 100] Throughout the manuscript please change the p-value to lower case to make the text consistent.

Reviewer 2 Report

This study established database of total sugar contents of yogurt product available in UK in 2019 and compared its reduction using the previous study in 2016. The topic is very interesting and well done. This type of field study is rare, but I believe that it is necessary to monitor the food products in the market place and evaluate government's action plan.

My comment is only one for organic products. In this study, it was emphasized that organic products had higher total sugar contents than non-organic products unexpectedly. However, as it was already described in the text, it is a matter of free sugar, not total sugar. As you said, yogurt is depend on the type of milk and the ingredients added during production. In general organic products tend to use higher proportion of milk and lower proportion of other ingredients that may led to increase total sugar intake, but does not increase free sugar. Thus, it can not say that higher total sugar contents in organic products are not good for health unless taking composition into account. Although limitation of not using free sugar in this study was mentioned, it is better to mention when to discuss the organic products, too. 

Reviewer 3 Report

Overall:

There is inadequate information provided regarding the database information and categorisation and definition of sugar in the methods section.

More information in the results section may be warranted – although it is not likely that ingredient level information can be supplied, information regarding common additives would be particularly useful – such as, lactose, fruit purees, etc. Without breakdown of sugar source, these information are essential for appropriate insight into the descriptive contents of the categories (perhaps arbitrary?) and the impact of the sugar content in the context of the food. I.e. is the trend that dairy content is being reduced? That fruit purees are being reduced? That actual added cane sugar is reduced? That non-nutritive sweeteners are being added (e.g. stevia?)?

The description of macronutrient composition differences between dairy categories does not seem relevant to the current investigation, and moreover no comparison of macronutrient content between 2016 and 2019 has been presented.

The results section is at times tedious and long, with repetition of data between figures and text.

The discussion should relate more closely to the results that have been presented – or the results should be reconsidered in light of the content provided in lines 282-309.

Additional discussion is suggested to encompass limitations in sugar definitions, as well as commentary on the reformulation or substitution practices underpinning these findings.

Abstract:

Why has the median and range been used? Why not the median and IQR? With 40% new products, what is the usefulness of using a paired comparison? Would it not be better to compare categories or nutritionally similar/similar marketing products/product brands rather than product SKU comparisons. Was the sugar content normalised to energy or serving size? Line 16 – minor typo Line 19 – extremely high level of precision?

Introduction:

Line 63 – 3.1% reduction in sugar intake does not sound substantial. Surely there are other food categories, similarly perceived as healthy foods, that would contribute more greatly – i.e. fruit juices. What is the specific rationale for focus on yogurt, if the proposed benefit is relatively insubstantial? Line 70 – in addition to the actual sugar content in yogurts, an important piece of information to interpret this study is the sales and consumption of each product/category of product.

Materials and methods:

Line 80 – the dates for the 2016 survey should also be indicated – was this January also? Line 79-80: what kind of bias in selection of omission of products does this online search result in? Are there regional differences in what is available online versus in store? Are there biases toward certain brands? Please provide comment on the risk of overrepresentation or omissions relative to availability in real life (as the majority of consumers will still be purchasing in person). Similarly, what kind of representation for supermarket shopping do these supermarket chains represent (e.g. 70%, 100%?). Search terms are limited to ‘yogurt, yoghurt’. Why is fromage frais mentioned in the introduction? Line 110 – please provide information about the source of the nutrient composition used in this study – were nutrient compositions taken from the online shopping sources? From manufacturers? From nutrient composition databases? Please provide comment on whether updates to the nutrient composition databases between 2016 and 2019 exist, and if so, the extent that these may have influenced the results. Importantly, more information about the definition of sugar in the nutrient composition needs to be provided – are ‘added sugars’ being counted as separate from total carbohydrate content? Or is total carbohydrate content counted as sugar? In the case of added whole fruit to yogurt – or complex carbohydrates such as oat or rice – do these count as ‘sugar’? What about lactose? Is this counted as sugar, carbohydrate, or both? Differences in this classification within databases or labelling between 2016 and 2019 could contribute substantially to this discrepancy. Line 112-119 – why were median and range used rather than median and IQR?

Results

Inconsistency of reporting significant digits Line 180 – how many of these sugar reductions can be attributed to the reporting of sugar? I.e. is sugar reported in terms of carbohydrate, or added sugar? Which is counted as sugar in the current survey? For instance, a greek unsweetened yogurt may have the same amount of lactose (i.e. the only sugar) in both 2016 and 2019 – is the change in sugar content due to a new label for lactose as “not added sugar”? and counted in the carbohydrate content rather than sugar content? Line 165-179 – these results are tedious, and would be better positioned in a table. Table 1 – units are missing, product descriptions are inadequate; I am unclear of the relevance of this table as it is not in comparison to the 2016 data. 227-229 – more appropriate for discussion; I also don’t see why this is surprising if it has been already reported previously. This whole paragraph should be limited to presentation of results rather than commentary.

Discussion

Line 246 – no macronutrient comparison with 2016 was made in the results. Line 283 – no analyses of consumer trends was made in the current study. Number of SKUs does not equal consumer behaviour. If anything, one might suspect that new SKUs are not at all representative of consumer behaviour if they have not yet proven themselves as stable in the market. 285-292 – what is the impact of alternative milk sources on the manufacturing of these products, particularly in relation to their carbohydrate and or sugar content? I would think that these products are actually more likely to have higher added sugar content than dairy based products. Commentary on this is required. Line 294-309 – really has nothing to do with the current investigation. Line 282-293; line 294-309: considerable air-time is given to market trends for new products in the yogurt category - -however, no comment of the sugar content in these products relative to traditional dairy products is made. Moreover, these sub-categories are not presented in the results section – if these are indeed important topics to discuss, then they should be adequately reported on in the results section. 310-326 – see comments above for missing limitations that should be discussed. The issue of lactose should be elaborated upon. As no information was presented on fruit purees in the current study – suggest either limiting this section of the discussion, or better yet, provide information in the results section about these practices in the UK products. The discussion lacks comment on the source of these reductions. The results indicated that sugar reductions were made without additional macronutrient substitution. Were the sugar reductions a results of non-nutritive sweeteners? Bulking agents? Water content? What were the changes in formulation, at least based on the NIPs? Or what is the required information in this area – shouldn’t we know what substitution ingredients we are therefore feeding to our children? Given the context of lines 327-336, this caution around the practices that achieve this should be made. If substitutions are substituting one NCD for other issues, then can we wholeheartedly support them? It would be nice to have some context on consumer behaviours of these categories, or an understanding of the source of these lower sugar products – are there specific brands that are leading the discontinuing of the high sugar products? Are specific brands remaining stagnant in their product and sugar availability?

Round 2

Reviewer 3 Report

I appreciate the authors thorough responses to the questions raised, and their effort to incorporate these clarifications within the manuscript for the reader. Given the differences/lack of information on food composition in some regions of the world, the additional information should make this paper highly relevant for non-UK or developping regions to consider evolving (and growing) dairy product availability and the implications for public health.

Minor typo: line 330 'manufactures' should read 'manufacturers'